# Type-Specific Seizure Network Analysis For RNS Target Selection in Multifocal Patients

Genchang Peng, Mehrdad Nourani[*]
*Department of Electrical and Computer Engineering*
*The University of Texas at Dallas*
Richardson, TX 75080
{Genchang.Peng, nourani}@utdallas.edu

Jay Harvey
*Department of Neurology*
*The University of Texas Southwestern Medical Center*
Dallas, TX 75390
Jay.Harvey@UTSouthwestern.edu

*Abstract*—**Responsive neurostimulation (RNS) is an effective device for patients with multifocal seizures whose ictal foci are independent of each other across left and right hemispheres. Accurate RNS placement is crucial to enhance seizure suppression outcomes. Stereo-electroencephalography (SEEG) is employed before RNS placement to collect deep brain activities and determine stimulation targets. To deal with different seizure types and semiologies for multifocal patients, this paper presents a functional seizure network model using SEEG to identify potential RNS targets. The network nodes are a subset of SEEG contact points, and directional weighted network edges are SEEG correlations quantified by directed transfer function (DTF). The network nodes are then ranked by their strength values, and the top-4 nodes are selected as RNS targets with respect to different seizure types in each hemisphere. The proposed methodology is validated based on five multifocal patients. Consistent results between our computational findings and clinicians' decisions are observed with 95.7% overlapping ratio.**

*Index Terms*—**Multifocal, responsive neurostimulation, seizure network, stereo-electroencephalography**

## I. INTRODUCTION

### A. Motivation and Background

Responsive neurostimulation (RNS) is a device and technology for epilepsy patients whose seizures are initiated from multiple seizure focus [1]. Traditional resective surgery, including open resection or laser ablation, cannot be performed to eliminate two or more seizure focus that are independently located across two hemispheres [2]. RNS is applied for seizure suppression by implanting one or two leads placed on different seizure focus to deliver pulse via contact points (4 contact per lead) [2]. These leads are connected with an implantable stimulator. When ictal events are detected, the stimulator automatically generates and transmit electrical pulses to the contact points to stimulate the seizure onset zones. For multifocal patients, whose ictal activities are independently originated from left and right hemispheres, RNS can achieve up to 53% seizure reduction rate after a 2-year observation period [3].

To achieve positive outcomes (practically $\geq 50\%$ seizure reduction rate), doctors need to carefully localize the seizure focus via different evaluation approaches. Stereo-electroencephalography (SEEG) is a minimally-invasive technique to collect brain activity data directly from the cortex via implantable depth electrodes [4]. Each electrode has 8-16 contact points for signal collection. When multiple SEEG electrodes are implanted across left and right hemispheres, neurologists need to visually inspect the hundreds of SEEG signals, and identify the stimulation targets (at most 8 points, 4 from each hemisphere). To automate the SEEG evaluation process and improve accuracy, a functional seizure network modeling is proposed [5]. The seizure network nodes are electrodes/contacts covering different brain regions. The network edges are functional connectivities denoting signal correlations. In particular, directed (or effective) connectivity metrics from various domains, such as time-domain Granger causality, frequency-domain directed transfer function and information-domain transfer entropy, are applied to characterize the *directional* connectomics of neural signals among different brain regions [6]. In a directional seizure network, nodes with strong outgoing influences are the initiating points that motivates seizure onset, and nodes with high incoming connections are the impacted points showing strong symptoms [7]. Various graph analytic tools are then applied to identify the nodal importance, and select the as stimulation targets [8].

While SEEG network analysis has been widely applied in selecting RNS targets for multifocal patients [8], [9], most existing works treat the ictal events equally from data learning perspective, and do not further take into account the variety of networking that can be seen across seizures in a single individual. With different clinical symptoms and ictal severity, e.g., focal aware seizures (with consciousness) versus focal impaired awareness seizures (without consciousness) [10], it is beneficial to adjust the stimulation settings (e.g., stimulating contacts, pulse parameters) with respect to each seizure type. This may prove further importance during the post-implantation period (e.g., 1 year), when physicians review the chronological record of seizure patterns and re-program the device to improve seizure reduction rate [11].

### B. Main Contribution

To assist clinicians evaluate the impact of RNS in suppressing seizures, and provide potential optimization for post-implantaion adjustment, we propose a type-specific seizure network model to identify RNS targets for different seizure events. Specifically, we select a subset of SEEG contact points as network nodes (i.e., spatial analysis), and use directed

---

[*] Corresponding author.

TABLE I: Clinical information of five study patients

| ID | Gender | Age of Implantation | Electrode No. (#) | Contact No. (#) | Seizure No. (#) and Type† | | Reduction Rate |
|----|--------|------|-----------|---------|---------------|----------------|------|
| | | | | | Left Hemisphere | Right Hemisphere | |
| 2000 | F | 39y | L: 11, R: 7 | 210 | 2 CPS, 4 SPS, 4 SUB | 4 SUB | 75% |
| 2200 | M | 37y | L: 11, R: 9 | 212 | 4 CPS | 3 CPS, 3 SUB | 25% |
| 2300 | F | 55y | L: 11, R: 5 | 182 | 5 CPS | 1 SPS, 1 SUB | 75% |
| 2400 | F | 37y | L: 10, R: 7 | 174 | 2 SPS, 6 CPS | 2 CPS | 25% |
| 2500 | M | 40y | L: 10, R: 11 | 252 | 6 SPS, 3 SUB | 1 CPS, 1 SUB | 50% |

† CPS: complex partial seizure; SUB: sub-clinical seizure; SPS: simple partial seizure

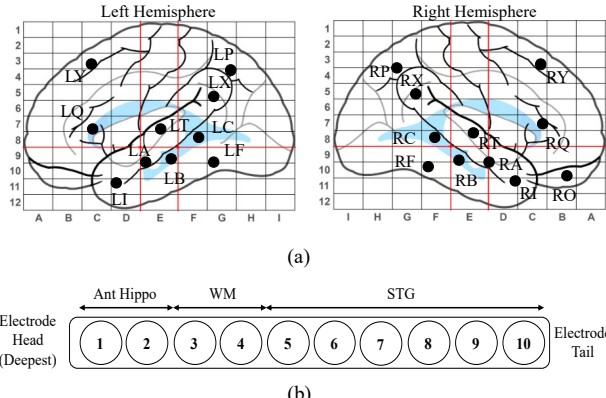

(a)

(b)

Fig. 1: Example of Patient 2500: (a) SEEG implantation map, (b) contact points from electrode LC. Ant Hippo: anterior hippocampus, WM: while matter, STG: superior temporal gyrus.

transfer function (DTF) as edge connectivity to characterize the interconnections during ictal activities (i.e., temporal analysis). We rank the network nodes based on their connectivity strength, i.e., sum of weights associated with incoming and outgoing edges, and select the top-4 points in each hemisphere. We compare the RNS target selection results for different types of focal seizures, and discuss the consistency with clinicians' choices.

This work is organized as follows. Background information are provided in Section II. Methodologies are presented in Section III. Experimental results are discussed in Section IV. Conclusions are in Section V.

## II. BACKGROUND

### A. Clinical Information

This study is conducted under Institutional Review Board (IRB) IRB-21-198 approved by the University of Texas at Dallas (UTD) and the University of Texas Southwestern Medical Center (UTSW). Five patients (two males and three females) were selected by a group of experienced clinicians, and their clinical information are provided in Table I. Patient inclusion is based on following criteria. First, these patients have failed to at least two different anti-epileptic drugs (AEDs). Second, each patient has two independent seizure onset zones from

left and right hemispheres. Specifically, all seizure events are marked as *focal-onset* by clinicians, meaning that their origins and network developments remain within the corresponding hemisphere [12]. Third, all these patients show different seizure types originated from at-least one hemisphere. Fourth, before RNS placement, all patients have undergone through SEEG implantation and evaluation in epilepsy monitoring unit (EMU). An example of SEEG implantation map for Patient 2500 is presented in Fig. 1(a), and the electrode contact information of LC electrode is shown in Fig. 1(b). SEEG signals were collected using Nihon Kohden EEG 1200 system at 1000 Hz sampling rate that are bipolarly referenced [13].

### B. Epilepsy & Terminology

In this work, epilepsy *seizure* refers to the recurrent ($\geq 2$ per day) brain abnormalities that are caused by the excessive neuronal misfirings [12]. In our study, there are three clinical seizure types annotated by experienced clinicians. We use the following annotations of seizure events according to the International League Against Epilepsy (ILAE) [10]:

**1. CPS:** Complex partial seizures (CPS) are the classical definition of focal seizures with loss of awareness. According to the latest definition [10], they are now referred as *focal impaired awareness seizures*.

**2. SPS:** Simple partial seizures (SPS) are those when patients do not lose awareness. According to the latest definition [10], they are now referred to as *focal aware seizures*.

**3. SUB:** Sub-clinical seizures (SUB) are the focal seizures or ictal abnormalities with no clinical symptoms explicitly observed (e.g., jerking). They are identified based on EEG discharges from clinicians' visual inspection.

In this study, all focal seizures of selected patients are confined within their onset hemispheres, and do not show secondary generalizations that propagate to the contralateral hemisphere (also known as focal to bilateral tonic-clonic seizures) [14]. All patients have experienced RNS treatment with at least 25% seizure reduction rate during the 1-year post-implantation period, as listed in Table I.

### C. DTF Seizure Network

Among various network connectivity metrics, directed transfer function (DTF), together with its multiple variants, has been widely applied for different tasks, including localizing

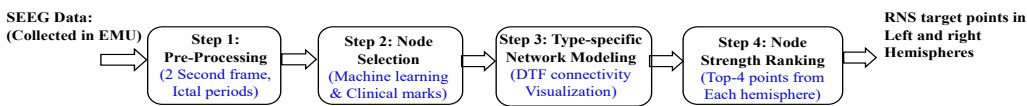

Fig. 2: Flowchart of the proposed methodology.

surgical zones [15], predicting seizure onsets [16] and recommending surgical options [17]. Established from multivariable auto-regressive (MVAR) model, DTF can better capture the spectral (rhythmic) components (e.g., $\beta$ band) of ictal activities than time-domain metrics, and are more computationally efficient than informational-based metrics [6]. Definition of DTF can be referred in [18]. For a standard $C$ variable MVAR with order $K$ (i.e., time lag):

$$\mathbf{x}(n) = \sum_{k=1}^{K} \mathbf{A}(k)\mathbf{x}(n-k) + \mathbf{e}(n) \qquad (1)$$

Its frequency-domain expression is written as $\boldsymbol{A}(f)\boldsymbol{x}(f) = \boldsymbol{e}(f)$. Then, we have:

$$\boldsymbol{x}(f) = \boldsymbol{A}^{-1}(f)\boldsymbol{e}(f) = \boldsymbol{H}(f)\boldsymbol{e}(f) \qquad (2)$$

Here, $\boldsymbol{H}(f) = \boldsymbol{A}^{-1}(f)$ is the $C \times C$ DTF transfer matrix. Each matrix element $h_{ij}(f)$ can also be viewed as the spectral coherence of frequency $f$ between $i$-th and $j$-th signals, capturing the correlation between the $j$-th input (column index) and $i$-th output (row index) of the system. The DTF value at frequency $f$ is given as [19]:

$$\phi_{ij}(f) = \frac{|h_{ij}(f)|^2}{\sum_{j=1}^{C} |h_{ij}(f)|^2} \qquad (3)$$

And the band DTF value $\phi_{ij}(f_1, f_2)$ over specific frequency band $[f_1 - f_2]$ is: [18]:

$$\phi_{ij}(f_1, f_2) = \sum_{f=f_1}^{f_2} \phi_{ij}(f) \qquad (4)$$

In this work, we chose $\beta$ band $(13 - 30\text{Hz})$ which has been widely studied to characterize seizure networking patterns [20], [21]. The initial choice of model order $K$ in Equation 1 is data-dependent, and can be optimized in a patient-specific (varied from 5 to 15) way using different criteria [18]. In this work, based on empirical recommendations and similar works [22], we choose $K = 10$ across all five patients. Personalization of parameter choices is beyond the scope of this work which remains for future investigation. Instead, we concentrate on the type-specific network modeling process, which will be illustrated next.

## III. RNS TARGET IDENTIFICATION METHODOLOGY

The basic flowchart of our proposed method has four steps as shown in Fig. 2. After collecting SEEG signals from multi-focal patients, we first pre-process the continuous signals into 2-second frames for data learning. Then in Step 2, we select network nodes from SEEG contact points. Step 3 is

a type-specific network modeling, which characterizes the spatial-temporal properties of seizure events using DTF edge connectivities. Step 4 is to rank network nodes based on their strength values. The results are the top-4 selected RNS points from left and right hemispheres.

### A. Data Pre-Processing

To focus on seizure events, in this work, we extract the ictal period (i.e., from clinical-marked onset to end) of seizure events from each patient. We divided the continuous SEEG signals into 2-second, $2 \times 1000 = 2000$ samples per frame, with $50\%$ lapping (1-second). A short frame (e.g., empirically less than 5-second) is appropriate to keep data stationary (quasi-stationary) for DTF estimation, and can result in more data segments for statistical analysis [7]. All signals are filtered with a band-pass filter at $[1, 60]$ Hz.

### B. Network Node Selection

SEEG implantation usually involves multiple electrodes with hundreds of contact points (see Table I). Clinical RNS devices requires at most 8 points for stimulation (4 points per hemisphere), which is the limit of existing device [9]. Also, since the seizure onsets are focally localized in a few areas, inclusion of less-contributing nodes will introduce ambiguities and impurities for the learning process. To build a practical network model with clinical explainability, in this study, we applied linear discriminant analysis (LDA) to rank out top-20 discriminant contact points as network nodes. This top-20 is empirically chosen based on clinicians' experiences and practical RNS requirements. As an effective filter-based feature selection method, LDA method has been widely employed for efficient channel selection in many SEEG studies [23]. Detailed calculation of LDA method can be found in [24]. Table II lists the top-20 LDA ranked points of Patient 2500 (10 left and 10 right), where the clinical-marked points (denoted in *) are included in the top-20 list, showing the consistency between our machine findings and clinicians' inspections.

TABLE II: 20 Contact points of Patient 2500 selected by LDA and clinicians' inspection

| Left Hemisphere | | Right Hemisphere | |
|---|---|---|---|
| Electrode | Contact Points | Electrode | Contact Points |
| LA | LA1, LA2* | RA | RA1, RA2 |
| LB | LB1*, LB2*, LB5, LB6* | RB | RB1*, RB2, RB4, RB6* |
| LC | LC1, LC2*, LC6, LC7 | RC | RC1*, RC2, RC6, RC7 |

∗ Clinical marked points

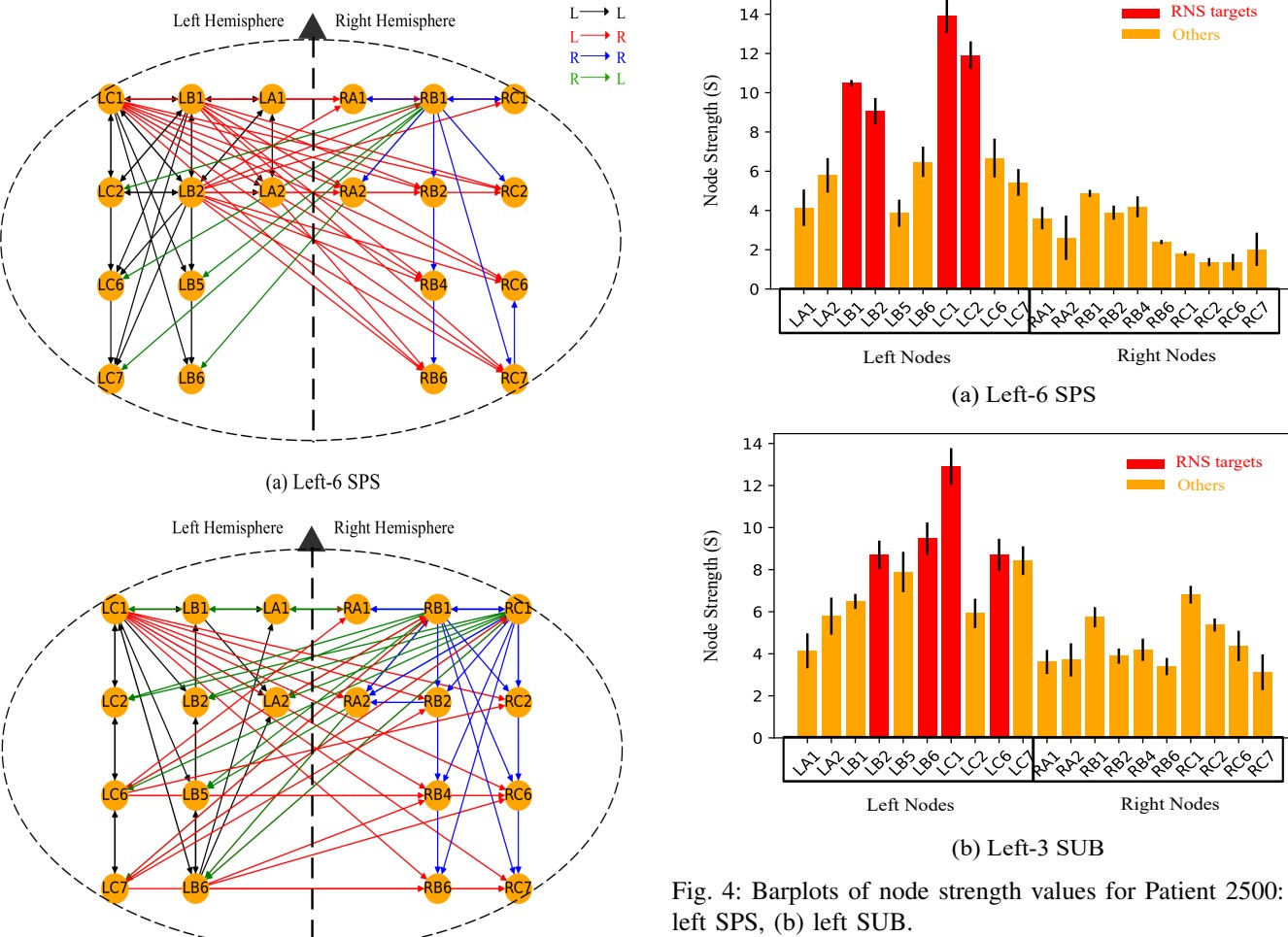

(a) Left-6 SPS

(b) Left-3 SUB

Fig. 3: Graph visualization of Patient 2500: (a) left SPS, (b) left SUB.

Fig. 4: Barplots of node strength values for Patient 2500: (a) left SPS, (b) left SUB.

## C. Type-Specific Network Modeling

After selecting $C = 20$ contact points as network nodes, we follow the procedure in Sec. II-C to assign DTF values as network edges, and construct the network model for each type of focal event. In Patient 2500, there are 6 SPS and 3 SUB events originated from left hemisphere (see Table I). For these two categories of focal events, we combine the seizure samples having the same annotations, and obtained the type-specific network models for (a) all left-6 SPS and (b) all left-3 SUB, respectively. Figures 3(a) and (b) visualize these two networks. Here the network nodes are the selected top-20 SEEG contact points (see Table II), and the edge connections (i.e., DTF values) between left (L) and right (R) nodes are differentiated via colors. To focus on the important connections, we discard the self-correlation, and maintain top-25% highest edge values for better visualization. It can be observed that, both SPS and SUB events have stronger left-originated connections (i.e., L→L and L→R) than the

right side (i.e., R→R and R→L), corresponding to their left initiations and developments. On the other hand, the distinction between left and right connectivities are highly discriminative in SPS network, but less discriminative in SUB network. Therefore, our network model captures the spatial-temporal patterns observed across different types of seizures.

## D. Node Strength Ranking

As a directional seizure network, each node has two edge connections: (i) outgoing edges and (ii) incoming edges. To measure the connections of $c$-th node within the seizure network, we define the node *strength* $S_c$ as the average DTF values of all outgoing and incoming edges linked to this node [5]:

$$S_c = \frac{1}{N}[\underbrace{\sum_{i=1,i\neq c}^{N} \phi_{i,c}}_{\text{outgoing}} + \underbrace{\sum_{j=1,j\neq c}^{N} \phi_{c,j}}_{\text{incoming}}] \qquad (5)$$

Here $\phi_{i,c}$ and $\phi_{c,j}$ are the outgoing ($c$-th column) and incoming ($c$-th row) DTF values from all $N$ points ($N = 20$ nodes), where self-correlation are excluded as marked in Equation 5). For the left-onset SPS and SUB events of Patient 2500, we

use barplots to show their node strength values in Figs. 4(a) and (b), respectively. Here the bar height is the mean (average) strength value per node, and the error bar shows the standard deviation by summing up across all events. The top-4 points with the highest strength values are marked in red, and are considered as our recommended RNS targets. For SPS events, the top-4 selection results include LB1-2, LC1-2, matching the clinical-marked onset points in Table II. SUB events, on the other hand, share LC1 and LB2 as commonly-selected results, but have LB6 and LC6 as specific findings. These observations indicate the importance of type-specific analysis, as the critical (e.g., triggering) nodes and developmental pathways (e.g., propagation) can be different with respect to each type of clinical seizure. Such knowledge may be utilized to change the stimulation settings of RNS.

### E. Validation

From data learning perspective, it is of interest to compare our type-specific results versus the *combined* results yielded by homogeneously analyzing all seizure types (i.e., type-independent). Using Patient 2500 as an example, for left and right hemisphere, the top-4 selected points of different seizure types are listed in Table III, where SPS & SUB for left and SUB & CPS for right. For each hemisphere, we combine the different seizure events together, and perform the node strength ranking to obtain the top-4 nodes as the combined RNS target. The common-selected points between type-specific and combined are marked in boldface. While there maybe some discriminants, in general, both the type-specific and type-independent results include the clinician-marked points (noted by *) as commonly-selected points, showing the consistency of identifying the initiating nodes in focal seizure networks. In real world scenarios, such consistency of combined analysis can help doctors to initialize the RNS stimulating points at the beginning, and the type-specific variances will be adopted for post-implantation adjustment during the observation period.

TABLE III: Type-Specific vs combined selection of Patient 2500

| Seizure | Target Selection Results | |
|---|---|---|
| Types | Left | Right |
| SPS | LB1, **LB2**\* **LC1**\*, LC2 | - |
| SUB | **LB2**\*, LB6 **LC1**\*, LC6 | **RB1, RB2**\* **RC1**\*, RC6 |
| CPS | - | **RB1, RB2**\* **RC1**\*, RC2 |
| Combined | LB1, **LB2** **LC1**, LC2 | **RB1, RB2**\* **RC1**\*, RC2 |

\* Clinician-marked points

To evaluate the validity of our approach, we compare our machine algorithm recommendation (ML) versus clinicians' decision (CL). Since our ML evaluation is applied for pre-implantation evaluation, we compare our top-4 combined re-

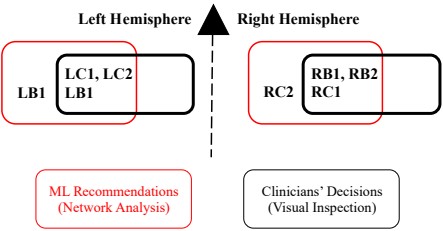

Fig. 5: ML-Based recommendation (red rectangle) vs clinicians' decision (black rectangle) for Patient 2500.

sults (for RNS initiating) versus clinicians' settings, as shown in Fig. 5. Using the top-4 ML selections (LC1, LC2, LB1 and LB2) of left hemisphere for example, 3 points are commonly selected by CL approaches (LC1, LC2 and LB2) in black. In this way, the *overlapping ratio* between two approaches is $\frac{|ML| \cap |CL|}{|CL|} = \frac{3}{3} = 100\%$, where we consider the clinicians' decision as standard reference.

## IV. EXPERIMENTAL RESULTS

### A. Summary of Type-Specific Selection

For the five Patients 2000, 2200, 2300, 2400 and 2500, we first show their type-specific RNS selection results in Figs. 6(a), (b), (c), (d) and (e), respectively. Using Patient 2000 for illustration, from left hemisphere, the commonly selected results of three seizure types are noted in black font (LB1, LB2 and LC1), and the additional selections from SPS, SUB and CPS events are differentiated in blue (LC2), green (LI1) and yellow (LC2), respectively. For the right hemisphere, where only SUB events are annotated, the top-4 selected points are our sole recommendations. Looking through these four cases, we observed that:

1) For each patient, the selection results of different seizure events (SPS, SUB or CPS) show overlaps within the same hemisphere, as marked in black font. The commonly-identified RNS targets correspond to the focal origins from each hemisphere, which can be the initial selections for RNS configuration.
2) When originated from identical hemisphere, each seizure event has its individual semiology, leading to alternative findings (differentiated via different colors). Our methodology can in generally help physicians to achieve optimum stimulation outcomes for their patients. Physicians could adjust the stimulation settings of targets based on the occurrence of specific seizure patterns.

### B. Comparison with Clinicians' Decisions

We have summarized our computational findings (global) and clinical decisions in Table IV. For each patient, we present the selection results from left and right hemispheres, and differentiate our machine learning selection (ML) and clinicians' decision (CL) by red and black box, respectively. To compare these two approach, we define the *overlapping ratio* of each hemisphere as $\frac{|ML| \cap |CL|}{|CL|}$. We note that:

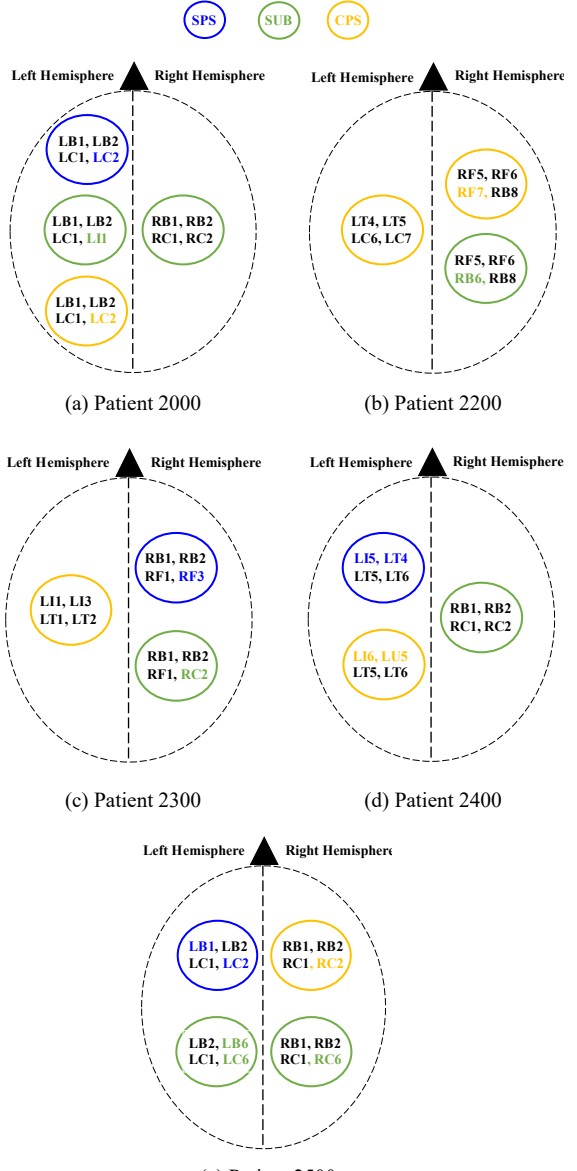

**Fig. 6:** Type-Specific RNS selection for different Patients: (a) 2000, (b) 2200, (c) 2300, (d) 2400 and (e) 2500. Here the commonly selected points are in black, and specific results of different types are marked in corresponding colors.

1) For all five patients, our top-4 selected points include clinicians' choices (less than 4 points), with overlapping ratio of 95.7% from left and right hemispheres. These indicate the consistency between two selection approaches (our machine selection and clinicians' decision). Especially, for cases where doctors selected 3 or less stimulation points (e.g., Patient 2000), they are all identified by our network analysis.

2) In the right hemisphere of Patient 2200, while RF5 and RF6 are selected by both approaches, the clinicians' additional choice is RB6, whereas our method's selection includes RF7 and RB8. We observe a similar scenario for

TABLE IV: Comparing selection results of five patients

| ID | Target Selection Results | | Overlapping Ratio (%) | |
|---|---|---|---|---|
| | Left | Right | Left | Right |
| 2000 | LC2 / LB1, LB2 LC1 | RC1 RC2 / RB1, RB2 | 100% | 100% |
| 2200 | LT4, LT5 LC6, LC7 | RF7 RB8 / RF5, RF6 RB6 | 100% | 66.7% |
| 2300 | LI3 LT1 / LI1 LT2 | RC1 RF1 / RB1, RB2 | 100% | 100% |
| 2400 | LT4, LU5 / LT5, LT6, LT7 | RB2 RC2 / RB1 RC1 | 66.7% | 100% |
| 2500 | LB1 / LC1, LC2 LB2 | RC2 / RB1, RB2 RC1 | 100% | 100% |
| Average: | ML Recommendation (Network Analysis) | Clinicians' Decision (Visual Inspection) | 95.7% | 95.7% |

the left hemisphere of Patient 2400. These two patients have 25% seizure reduction rate lower than other cases. We hypothesize that, our extra findings could have shown further possibilities to improve the outcomes.

## V. CONCLUSION & FUTURE DIRECTIONS

Multifocal patients demonstrate ictal EEG patterns initiating from more than one site and may have different seizures with specific patterns and semiologies (e.g., focal aware vs focal impaired awareness) in each hemisphere. Our seizure network analysis shows both consistency and alternative for the selection of RNS points. We use directed transfer function to measure the SEEG connections among node points, and rank the top candidate points of different seizure types by their strength values. All five studied patients have well-focalized seizures, leading to the consistent results of selection results. The alternatives for RNS targets enhance clinicians understanding of different seizure semiologies. Particularly, our network analysis can: (a) first recommend the initial stimulation targets (combined), and (b) assist clinicians configure RNS devices when specific seizure patterns are more frequently observed on the long run (type-specific).

In future, we will include a larger cohort of patients, especially for patients that have focal to bilateral tonic-clonic seizures and generalized seizures. Also, to evaluate the effectiveness of network analysis, patients whose RNS treatment are not successful should also be studied. The impacting factors that affect neurostimulation outcomes need to be investigated as well.

ACKNOWLEDGEMENT

This project was partially funded by The University of Texas at Dallas Office of Research and Innovation through the SPARK Grant Program.

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
