# OpenReview forum: "Type-Specific Seizure Network Analysis For RNS Target Selection in Multifocal Patients"
_IEEE.org/EMBS/BHI/2024/Conference — IEEE BHI'24_

### Official Review · Reviewer_uvwT · 2024-08-08
**The demonstration of automatic RNS target selection using the proposed sEEG network analysis is encouraging.**

**Overall Rating:** 7
**Confidence:** 3

**Other Quality Metrics:**

(a) Clarity of writing: good
(b) Clinical Significance: good
(c) Methodological Novelty: good
(d) Experiments and Results: good

**Questions For The Authors:**

1. This study argues that previous research did not focus on type-specific network analysis. Therefore, it is recommended that this study measures the consistency for each type separately. The subsequent comparison between type-dependent and type-independent network analysis outcomes will surely address this argument.

**Strengths:**

1. Unlike previous studies, this study demonstrates the capability to identify RNS targets corresponding to different seizure types in multifocal patients. Such type-specific selection outcomes potentially help clinicians to adjust the RNS parameters to improve the seizure reduction rate.
2. This study verifies the methodological effectiveness by measuring the consistency between machine algorithm recommendations and clinicians' decisions.

**Summary Of The Paper:**

This study proposes a DTF-basis network analysis using sEEG to automatically identify the top-4 contacts per hemisphere as RNS targets, which is highly consistent to clinicians' decisions (~ 94% overlapping ratio).

**Weaknesses:**

1. The DTF algorithm employed in this study highly relies on the choice of its critical parameters, including model order, time segment, and node number. The authors are encouraged to justify how these parameters are objectively optimized and how they potentially modulate the RNS target selection consistency.
2. Have the authors compared the DTF results based on surrogate data? Due to volume conduction, EEG signals between neighboring channels/contacts are highly correlated, introducing inherent connectivity strength. A statistical comparison between surrogate and real connectivity helps to justify whether the revealed connectome truly reflects seizure-related associations.
3. It is not clear why LDA has to be applied to trim down the number of nodes for network analysis. Do the authors construct a DTF network in the original node space? What is the difference in the consistency of RNS target selection?

---

> ### Author Rebuttal · Authors · 2024-09-03
>
> We appreciate the reviewer’s time and suggestions. Our responses are concise due to limited spaces.
>
> Weakness 1: We have highlighted our explanation in Section II C. The first parameter of model order K is chosen as K=10 based on the recommendation from previous work [1]. The second parameter is frequency range, which can be defined between any range. Here we chose beta band (13,30 Hz) as it is widely applied in EEG seizure analysis. For the length of time segment W, we choose W=2 sec. so that each time frame can correspond to the observational intervals within the clinical annotations. Note that, the optimal parameters are determined based on datasets, which vary from patients to patients. As an investigatory work on seizure network modeling and RNS target selection, we showed our methodology yields consistent results in a seizure-specific style with relevant findings. Patient-specific optimization of DTF parameters (e.g., band) will be included in our future investigation.
>
> Weakness 2: We appreciate your instructive suggestions! We have not done it yet. Surrogate data is effective in removing the spurious correlations during edge connectivity estimation, especially when such spurious correlations are caused by anatomical distances [2]. In our network modeling, by LDA selection, we identify the top-20 nodes that are sparse in the graph’s nodal space. These discriminant nodes, which shows strong discriminant, are the potential RNS candidates to reflect the connectomics of EZ networks. Therefore, we estimate the DTF based on real-world SEEG data. In future, when larger graph space is explored (e.g., 50-150 nodes) to characterize the complexity of networking, we will incorporate surrogate testing to enhance the statistical robustness of our DTF estimation.
>
> Weakness 3: We have highlighted our rationale in Section III B. While it is computationally feasible to include the entire SEEG points (100-200) as graph’s search space, we apply LDA to choose top-C (C=20 here) nodes due to the following factors. First, practical RNS devices with FDA approval for surgery require no more than 8 points (max 4 per hemisphere [3]). Second, in multifocal patients, especially those with well-concentrated focal seizures, the ictal initiations are usually well-localized in a few discriminant points. Including too many other less-impactful points will introduce impurity and ambiguity for data learning. Therefore, we apply LDA to pre-filter the potential (highly impactful) candidate nodes.
>
> Question 1: We compare and discuss the consistency between type-specific vs combined (type-independent) results in Table III. Please refer to the highlighted part of Section III E in our revised draft.
>
> References:
>
> [1] C. Wilke, L. Ding and B. He, "Estimation of Time-Varying Connectivity Patterns Through the Use of an Adaptive Directed Transfer Function," IEEE Transactions on Biomedical Engineering, vol. 55, no. 11, pp. 2557-2564, Nov. 2008.
>
> [2] G. Chiarion, L. Sparacino, Y. Antonacci, L. Faes, L and L. Mesin, “Connectivity Analysis in EEG Data: A Tutorial Review of the State of the Art and Emerging Trends,” Bioengineering 2023, 10, 372.
>
> [3] T. L. Skarpaas, B. Jarosiewicz, and M. J. Morrell, “Brain-Responsive Neurostimulation for epilepsy (rns® system),” Epilepsy Research, vol.153, pp. 68–70, 2019.

---

### Official Review · Reviewer_F5Fe · 2024-08-08
**This paper presents a novel approach for selecting responsive neurostimulation (RNS) targets in multifocal epilepsy patients by developing type-specific seizure network models. The method uses directed transfer function (DTF) to analyze stereo-electroencephalography (SEEG) data and identify critical nodes for RNS targeting. The study is valuable for its potential to improve RNS outcomes by tailoring interventions to specific seizure types, but some methodological and clinical aspects could be further strengthened.**

**Overall Rating:** 5
**Confidence:** 4

**Other Quality Metrics:**

Clarity of Writing: Good.
Clinical Significance: Good.
Methodological Novelty: Fair.
Experiments and Results: Fair.

**Questions For The Authors:**

1. How does your method compare to other existing machine learning or statistical approaches for RNS target selection in terms of accuracy and clinical outcomes?
2. Your study argues that previous research did not focus on type-specific network analysis. Can you measure the consistency of RNS target selection for each seizure type separately and compare the outcomes between type-dependent and type-independent analyses to support this argument?
3. How were the critical parameters in the DTF algorithm optimized, and how might these parameters influence the robustness of your findings?

**Strengths:**

1. The study addresses a clinical need by enhancing RNS target selection for multifocal epilepsy patients. Unlike previous studies that treat seizures homogeneously, this study demonstrates the capability to identify RNS targets corresponding to different seizure types in multifocal patients. Such type-specific selection outcomes potentially help clinicians adjust RNS parameters to improve the seizure reduction rate, which is a novel and clinically valuable contribution.
2. The use of DTF for constructing type-specific seizure networks is innovative, allowing for a more nuanced understanding of seizure dynamics across different types of seizures.
3. The study effectively validates the methodological approach by measuring the consistency between the machine algorithm's recommendations and clinicians' decisions, achieving a high overlap (94.4%). This validation supports the reliability of the proposed approach in a clinical setting.
4. The method is data-driven and uses objective SEEG data, reducing potential biases in target selection compared to purely clinician-based approaches.

**Summary Of The Paper:**

The paper introduces a method for selecting RNS targets in multifocal epilepsy patients using type-specific seizure network analysis. It uses stereo-electroencephalography (SEEG) data to construct functional networks where nodes represent SEEG contacts, and edges represent directional connections quantified by directed transfer function (DTF).
By ranking nodes based on their connectivity strength, they identifies the top-4 nodes in each hemisphere as potential RNS targets. They validates the approach on four patients, showing a high overlap (94.4%) between the computational recommendations and clinicians’ decisions, suggesting the method's utility in clinical practice.

**Weaknesses:**

1. The paper does not provide a detailed comparison with other existing methods beyond clinician selection which is quite subjective, such as other machine learning or statistical approaches for target selection (even from other disease application is acceptable). This limits the understanding of how much the proposed method improves upon current practices.
Specifically, the use of Directed Transfer Function (DTF) in seizure network modeling, particularly with stereo-electroencephalography (SEEG) data, is not unique to this study. DTF has been applied in various studies to analyze functional brain connectivity for different purposes, including seizure prediction and identifying epileptogenic zones. For instance, DTF has been used in conjunction with machine learning models like convolutional neural networks (CNNs) to predict seizures and analyze brain network dynamics in epilepsy patients​ (PubMed: https://pubmed.ncbi.nlm.nih.gov/33147147/).
While the method is well-established, the specific application and combination with other techniques (the type-specific network modeling in multifocal epilepsy) would determine the novelty of the current study. Regarding this, further comparisons with existing methodologies would clarify the unique contributions of this work, including how this method is applied or combined with other techniques, such as in the specific context of multifocal epilepsy or the use of type-specific seizure networks.

2. The study is validated on only four patients, thus limiting generalizability of the findings. More extensive validation on a larger and more diverse cohort is necessary. Moreover, the author mentioned different subtypes of epilepsy, is this method also practical on other types.

3. While the paper introduces the use of DTF for network construction, it lacks detailed discussion on why DTF was chosen over other potential metrics or methods for analyzing functional connectivity, such as Granger causality or coherence analysis. Also, how to adjust the parameters they used in the study need to be clarified.

5. Linear Discriminant Analysis (LDA) is used for node selection, but the paper does not sufficiently justify why LDA is applied to reduce the number of nodes before network analysis. The authors should explain whether DTF networks are constructed in the original node space and compare the consistency of RNS target selection with and without LDA filtering.

---

> ### Author Rebuttal · Authors · 2024-09-03
>
> We appreciate the reviewer’s suggestions. Our responses are concise due to limited spaces.
>
> Weakness 1: SEEG datasets are highly patient-specific, and the analytic approaches are task-dependent. We used private data collected at our affiliated institution. Currently, there is no public domain dataset that we can try. So, we cannot directly compare our RNS selection results with any existing work mentioned by reviewer, e.g., Ref [1] for seizure prediction task. Our work is novel for the SEEG data analysis that leads to assistive RNS decision making in a seizure-specific analysis framework.
>
> Weakness 2: We have added another patient case (Patient 2300). Due to our patient selection criteria (highlighted in Section II A), SEEG samples from multifocal patients that we could access are quite limited at this time. For seizure subtypes, our preliminary investigation includes patients having CPS, SPS and SUB, which are the common types of focal seizures defined in Ref [2] (highlighted in Section II B). In future, we will include more cases whose RNS outcomes are lower (i.e., 50%) due to focal to bilateral tonic-clonic (F-GTC) seizure, or generalized seizures.
>
> Weakness 3: We have improved our rationale on choosing DTF in Section II C. There are two key parameters in DTF. The first parameter is model order K, which is chosen as 10 based on the recommendation from previous work [3]. The second parameter is frequency range, which can be defined between any range. In our work, we chose Beta band (13,30 Hz) as it is widely applied in EEG seizure analysis. Please note that, the optimal parameters are determined based on datasets, which vary from patients to patients. As an investigatory work on seizure network modeling and RNS target selection, we showed our methodology yields consistent results with type-dependent findings. Patient-specific optimization of DTF parameters (e.g., model order, frequency band) will be explored in our future investigation.
>
> Weakness 4: We have highlighted our explanation in Section III B.  While it is computationally feasible to include the entire SEEG points (100-200) as node space, we apply LDA to choose top-C (C=20 here) nodes due to the following factors. First, practical limitation of RNS devices (approved by FDA and available for implantation) requires no more than 8 points (each hemisphere at-most 4 [4]). Second, in multifocal patients, especially those with well-concentrated focal seizures, the ictal initiations are usually well-localized in a few discriminant points. Including redundant and less-impactful points will introduce impurity and ambiguity for data learning. Therefore, we apply LDA to pre-filter the highly relevant candidate nodes.
>
> Question 1: Note that the SEEG data are highly patient-dependent. To our knowledge, there is no SEEG data and annotation (e.g. patient's clinical information, seizure types) collected from multifocal patients in public domain.  Our team is the first who analyzed seizure network in a type-specific way for RNS target selection. So, we cannot directly compare our results with any existing work reported in literature. This is a novel work that provides new vision on SEEG data analysis and assistive RNS decision. Besides, methodologies involved in different works are task-dependent, including machine learning scheme, evaluation standards, etc., that are incomparable with each other.
>
> Question 2: In Section III E, we compare and discuss the consistency between type-specific vs type-independent results in Table III.
>
> Question 3: We have highlighted our explanation on DTF parameters in Section II C. The model order K is chosen as K=10 based on the recommendation from previous works [3]. Second is frequency range, which can be defined between any range. Here we choose beta band (13,30 Hz) as it is widely applied in EEG seizure analysis. Note that, the optimal parameters are determined based on datasets, which vary from patients to patients. Patient-specific optimization of DTF parameters will be our future investigation.
>
> References:
>
> [1] G. Wang, D. Wang, C. Du C, K. Li, J. Zhang J, Z. Liu, Y. Tao, M. Wang, Z. Cao and X. Yan, “Seizure Prediction Using Directed Transfer Function and Convolution Neural Network on Intracranial EEG”, IEEE Trans Neural Syst Rehabil Eng. 2020 Dec;28(12):2711-2720. doi: 10.1109/TNSRE.2020.3035836. Epub 2021 Jan 28. PMID: 33147147.
>
> [2] J. J. Falco-Walter, I. E. Scheffer, and R. S. Fisher, “The New Definition and Classification of Seizures and Epilepsy,” Epilepsy Research, vol. 139, pp. 73–79, 2018.
>
> [3] C. Wilke, W. Van Drongelen, M. Kohrman, and B. He, “Neocortical Seizure Foci Localization by Means of a Directed Transfer Function Method,” Epilepsia, vol. 51, no. 4, pp. 564–572, 2010.
>
> [4] T. L. Skarpaas, B. Jarosiewicz, and M. J. Morrell, “Brain-Responsive Neurostimulation for epilepsy (rns® system),” Epilepsy Research, vol.153, pp. 68–70, 2019.

---

### Decision · Program_Chairs · 2024-09-23

Accept